# Examining Final-Administered Medication as a Measure of Data Quality: A Comparative Analysis of Death Data with the Central Cancer Registry in Republic of Korea

**DOI:** 10.3390/cancers15133371

**Published:** 2023-06-27

**Authors:** Yae Won Tak, Jeong Hyun Han, Yu Jin Park, Do-Hoon Kim, Ji Seon Oh, Yura Lee

**Affiliations:** 1Department of Information Medicine, Asan Medical Center, University of Ulsan College of Medicine, Seoul 05505, Republic of Korea; yaewon.c.tak@amc.seoul.kr (Y.W.T.); jhhan121@amc.seoul.kr (J.H.H.); doogie55@naver.com (J.S.O.); 2Medical Information-Management Team, Asan Medical Center, Seoul 05505, Republic of Korea; pyj81@amc.seoul.kr; 3Medical Big Data Research Center, Kyungpook National University Hospital, Daegu 41944, Republic of Korea; k8016851@gmail.com

**Keywords:** death, mortality, data quality, public data, data accuracy

## Abstract

**Simple Summary:**

Death represents the definitive endpoint for a patient; therefore, it is crucial to determine an accurate date of death. This study aims to examine the final-administered medication in a gold standard cohort that assesses death data accuracy. By utilizing electronic health records from a single medical institution and the Korean Central Cancer Registry, we establish the gold standard as patients who died in the hospital after the implementation of electronic health records, with a difference of 0 or 1 day between the final hospital visit/discharge and death. We calculate the similarity of the terminal medication between the gold standard and cohorts using cosine similarity. The findings reveal a positive correlation between mortality rates and similarities of the final-administered medication. This study introduces the potential of the last administered medication as a novel data quality measure of death data when the date of death differs between datasets.

**Abstract:**

Death is a crucial outcome in retrospective cohort studies, serving as a criterion for analyzing mortality in a database. This study aimed to assess the quality of extracted death data and investigate the potential of the final-administered medication as a variable to quantify accuracy for the validation dataset. Electronic health records from both an in-hospital and the Korean Central Cancer Registry were used for this study. The gold standard was established by examining the differences between the dates of in-hospital deaths and cancer-registered deaths. Cosine similarity was employed to quantify the final-administered medication similarities between the gold standard and other cohorts. The gold standard was determined as patients who died in the hospital after 2006 and whose final hospital visit/discharge date and death date differed by 0 or 1 day. For all three criteria—(a) cancer stage, (b) cancer type, and (c) type of final visit—there was a positive correlation between mortality rates and the similarities of the final-administered medication. This study introduces a measure that can provide additional accurate information regarding death and differentiates the reliability of the dataset.

## 1. Introduction

Death is a fixed outcome measure in retrospective cohort studies; therefore, the date of death is fundamental to analyzing mortality owing to its coherent ability to mark the end of observation [1]. Mortality is a more informative endpoint than disease-specific death since it avoids issues such as patient-selection bias, data absence, and temporal classification adjustments [2]. Considering that death is an inevitable termination point of diseases, building a database with high-quality mortality information is crucial as the length of survival is a vital indicator of prognosis. Data with reliable mortality could be used for research, health planning, and decision-making [3]. In Republic of Korea, mortality data are collected at the national level by the Office for National Statistics (ONS). 

The incoherence in death data can affect mortality statistics, which could lead to significant consequences such as low data reliability and later inaccurate research results. Many US insurance companies assert that deaths outside the hospital are unlikely to be recorded, resulting in only death at discharge being recorded in the database [1]. In the United Kingdom, one-fifth of patients had a date of death recorded at a general practitioner practice that was later than the date recorded by the ONS, which is based on the official death certificate [2]. 

The incidence of cancer has increased ceaselessly; therefore, the cumulative data regarding cancer mortality has surged as well. Some studies have demonstrated that clinical data in electronic health records (EHR) can be employed to identify false-positive (identified dead but alive) matches to ameliorate the accuracy of both clinical and research databases [4]. When the UK primary care data from the Clinical Practice Research Datalink were matched with the ONS data, the dates of death in primary care records often mismatched the national records [5]. Furthermore, in tertiary hospitals, the normal clinical path to death could be unavailable because of the priority regarding acute patients. Guiding patients to hopeless discharge or hospice care for a more peaceful departure often leads to the absence of tertiary hospital mortality data [6]. 

Therefore, previous research has evaluated the quality of mortality data with various methodologies. Data quality of vital statistics in death data was assessed using six dimensions, which included quality of cause of death reporting, completeness of death reporting, and more [7]. Furthermore, other researchers have assessed data quality for the cause of death in various countries’ mortality datasets depending on statistical models [8]. Moreover, there has been a retrospective analysis of the World Health Organization mortality data, which assessed age-adjusted unintentional fall mortality [9]. As the evaluation of the mortality data mainly relies on the cause of death, retrospective research to determine the deceased patients based on death dates in the database is warranted.

This study aims to evaluate the quality of the extracted death dataset using death dates and determine the potential of using final-administered medication as a measure to quantify the accuracy of the validation dataset. This study is the first to examine the accuracy of death dates of individual EHR. The survival of cancer patients can be affected not only by cancer progression but also by treatment complications, sequelae, and unexpected events, whereas death is a stable indicator of the course of the disease. Therefore, the effort to pursue accurate death information is worthwhile, indicating the importance of creating methodologies to determine the accuracy of a dataset with death-related information. Governing the accuracy of death datasets could benefit practitioners and suggest a novel method to identify cancer progress by examining the administered medication without complicated tests.

## 2. Materials and Methods

### 2.1. Overview

Using extracted death records from a single medical institution and the national cancer registration system, we compared patients’ death dates to determine the accuracy of the death dates in the datasets. Moreover, using the cosine similarity of the patients’ final-administered medication, formulated from the determined dataset gold standard (GS), we evaluated the potential of final-administered medication as an accuracy measure of death datasets. 

### 2.2. Data Sources

The Asan Medical Center (AMC) is the largest tertiary hospital in Republic of Korea, with 2715 beds, which ensures a large database that cumulatively includes data from over 4.3 million patients. After implementing the Order Communicating System in 1989, an in-house EHR system was implemented in 2007. Since 2006, death information has been computerized and managed in the hospital information system. Through the Korean government’s introduction of the Personal Information Protection Act and the Bioethics and Safety Act in 2013, the AMC has created a clinical data warehouse for a de-identified hospital information system that includes EHR [10]. The stored EHRs were accessible through the Asan Biomedical Research Environment (ABLE) service offered by the Asan Medical Information System, a pseudonymized clinical data warehouse. 

Demographic data, including death-related data in Korea, have been gathered by the ONS since 1978. Figure 1 provides a visual representation of the process. The process for data collection first begins at each medical institution, which registers cancer patients [11]. Then, each patient is registered at the Central Cancer Registration Headquarters. Following this, the Central and Regional Cancer Registration Headquarters conduct medical record investigation, data collection, and refinement of missing person data. Finally, the Central Cancer Registry analysis of the processed data is sent to the Ministry of Health and Welfare [11]. 

The Korean Central Cancer Registry (KCCR) is an official central registry of the Ministry of Health and Welfare, which has published annual cancer statistics reports for estimating patients’ survival and disease prevalence in Korea since 1980 [11]. The KCCR database comprises additional medical record review surveys, population-based regional cancer registries, site-specific cancer registries, and death certificates [11]. Cancer cases diagnosed in hospitals nationwide during the previous year are collected in the database at the Central Cancer Registry [13]. To ensure accuracy and completeness, the KCCR collaborates with central and regional cancer registries to compile data [13]. The cancer registration data distributed by the KCCR includes data such as first-time diagnosis date, cancer treatment information, cancer stage, surgical histological information (TNM classification), and death information. All cancer cases were registered based on the International Classification of Diseases for Oncology, 3rd edition (ICD-O-3) [13]. Moreover, the tumor stage is set within 4 months of diagnosis, as documented in the medical records, which is later captured by the KCCR [14]. These cancer-related data were obtained from the ABLE system.

In this study, the clinical data were obtained from the ABLE system as it has access to data regarding in-hospital and cancer-registered deaths from the KCCR. The study population comprised patients with a history of death in the AMC after 2006. The data included the EHRs of patients from both the AMC and KCCR with a death date between June 1989 and January 2021. It was acknowledged that there was a possibility that some patients’ deaths were not included in the study population due to the time lag between the KCCR cancer registry information being updated to the ABLE hospital system. Therefore, there were 179,747 selected patients for this study, with 125,819 patients with a date of death after 2006 (Figure 2). Cancer-registered death was designated as data with KCCR-only identified death, while in-hospital death was specified as the remainder (Figure 2).

### 2.3. Surveillance, Epidemiology, and End Results Stage

Surveillance, Epidemiology, and End Results (SEER) stage information is highly sensitive, with a high positive predictive value for cancer; therefore, it was used to elaborate the research [15]. The SEER stage information is entered when the cancer registration data from the KCCR is registered in the hospital. In the SEER stage, the eight categories [16] were re-categorized into five new categories for this study, and 0 and 1 were maintained but renamed as “In situ” and “Localized”, respectively. SEER stages 2, 3, and 4 were grouped as “Regional” because of their common regional characteristics. SEER stage 7 was renamed “Distant” because, in this stage, distant metastases have occurred. Finally, SEER stages 5 and 9, were designated as “Unknown” because, in these stages, it is unknown whether there was further spread of metastases. 

### 2.4. Anatomical Therapeutic Chemical Classification System

The Anatomical Therapeutic Chemical (ATC) classification system classifies drugs into separate groups based on an organ or system and their therapeutic, pharmacological, and chemical qualities [17]. The ATC classification system consists of several levels. The 1st-level ATC class has 14 main anatomical or pharmacological groups expressed by letters [17]. In the 2nd-level ATC class, the group is again categorized into its pharmacological or therapeutic subgroups expressed in double-digit numbers [17]. 

### 2.5. Cosine Similarity

Cosine similarity is a standard approach used to compute the similarities between two given vectors using the vector space model [18]. To calculate the cosine similarity between the two groups, the final administered drug is grouped by linking it to the ATC code, with each cell containing the sum of the count of each 2nd-level ATC class, as presented in Table 1.

Next, each value is transformed into its vector form as indicated:$$\alpha =[\begin{array}{cccccc} \alpha_{1} & \alpha_{2} & \alpha_{3} & \alpha_{4} & \alpha_{5} & \ldots \;\alpha_{n} \end{array}]$$
$$\beta =[\begin{array}{cccccc} \beta_{1} & \beta_{2} & \beta_{3} & \beta_{4} & \beta_{5} & \ldots \;\beta_{n} \end{array}]$$

Then, a cosine value is calculated according to the following formula:$$\mathrm{cosine}\;\mathrm{similarity}(\alpha ,\beta )=\frac{\alpha \cdot \beta }{\left\|\alpha \right\|\cdot \left\|\beta \right\|}$$
where vector α represents the 1st-row values from the table, reformatted as a vector value, and vector β represents the 2nd-row values from the table, reformatted as a vector value. α·β indicates the product between vectors α and β, and $\left|\left|\alpha \right|\right|$ represents the length of vector α. This measure represents the similarity between two vectors with a positive range from 0 to 1, where the near 0 value suggests low similarity, and near 1 suggests high similarity. The data analysis was performed using Python (ver. 3.8.5). 

### 2.6. Validation Dataset

The validation dataset included patients for whom it was possible to obtain cancer registration data from the national cancer registration project with a date of a cancer diagnosis from January 2008 to December 2009. Initially, there was a total of 23,512 accumulated patients. Using three exclusion criteria, which included cancer-registered death error, first visit date error, and unknown cancer stage (SEER stage of 5 or 9), the validation dataset was reduced to 18,909 (Figure 3). The final selected validation dataset had a death percentage of 42% (7943/18,909). 

Therefore, three criteria (a) SEER stage, (b) cancer type, and (c) type of final visit were chosen to determine whether cohorts created through these three criteria with different mortality would result in different cosine similarity to the GS, indicating a relationship between the patients’ mortality and the measure. To show the linear relationship between mortality and similarity, an R-squared regression analysis was conducted, and the R-squared value was calculated to show how fitted the best-fit line is. Therefore, we evaluated the accuracy of the extracted data for different variables by extracting deceased patients’ data. According to the source or management environment of the information on death, each variable’s reliability was determined. Consequently, the accuracy of death information was quantified using additional information associated with death, such as the final visit date and medication information. Moreover, by employing the information from the validation dataset, we determined whether the use of the final-administered medication could be a measure of death data quality. The all-cause mortality rate of each dataset was calculated considering the ratio between the number of people with non-cancer-specific deaths and the full population of the dataset.

## 3. Results

### 3.1. Date of Death Data Quality Differences

The yearly distribution of cancer-registered and in-hospital deaths is shown in Figure 4, and the general characteristics of 179,747 deaths are presented in Appendix A. Because the computerized management of the Hospital Information System started in 2006, the possibility of duplicated registrations of hospital and cancer-registered deaths from the KCCR has been significantly reduced; hence, the sharp decline in the number of in-hospital deaths. Moreover, because the trends in medication prescribed could cause differences, we deemed it suitable to classify the data based on before/after 2006. Consequently, the extracted data were initially classified into the following three categories: (a) Deaths before 2006, (b) In-hospital deaths after 2006, and (c) Cancer-registered deaths after 2006 (Figure 2). 

First, for the determined dataset, the differences between the in-hospital death dates and the cancer-registered death dates were reviewed. Most individuals (14,278/24,229, 58.9%) with in-hospital deaths after 2006 had accurate dates. Only a fraction (90/24,229, 0.371%) had inaccurate death dates between the in-hospital and cancer-registered dates. Moreover, less than half (9861/24,229, 40.7%) had an in-hospital death date but no cancer-registered death date. 

To be certain that the deceased patient died, a 60-day period was selected for the date difference between the last hospital visit and the death date. This is because of the possibility of false reports; the trustworthiness of the data decreases immensely 60 days post the death date of clinical activity [19].

To validate the accuracy between the in-hospital and cancer-registered death dates, patients with both in-hospital and cancer-registered death dates were compared. There were 326 (326/179,747, 0.18%) patients whose two dates did not match completely. Hence, after excluding the dates that were 1 day apart between the in-hospital and cancer-registered death dates, due to delayed processing because of office hours, there were 179 (179/179,747, 0.099%) patients whose two dates did not match. The cancer-registered death, recorded at the ONS, is released to the medical institution through the clinical research data warehouse, where the process is computerized, suggesting the low possibility of human error. Therefore, the point where human error could most likely occur is when the hand-written death report is computerized because of the complexity of EHR and the vast quantity which could include data entry mistakes, data selection errors, and data conversion errors [20].

As the possibility of false reports and untrustworthy data increases 60 days post the death date [19], the patients with 60 days or more difference between their in-hospital and cancer-registered death dates were 46 (46/179,747, 0.026%). We reviewed individual EHRs for 19 (19/179,747, 0.011%) patients with differences between in-hospital and cancer-registered death dates exceeding one year. For all 19 patients, the in-hospital death dates were accurate, with direct or indirect records that indicated expiration. Therefore, to confirm which death date was more reliable, the patients with more than a 1-year difference were chart reviewed. We confirmed that in-hospital deaths were more accurate than cancer-registered death because the death record from in-hospital deaths had a death-related record or a circumstance related to death, such as “post-mortem care”. Thus, we could assume that the date of death in the hospital was more likely to be accurate than the cancer-registered death.

### 3.2. Gold Standard

Through examining the death dates, we were able to conclude that the cohort with the highest accuracy of death date would be the in-hospital patients who died after 2006 with a 0- or 1-day difference between their final hospital visit/discharge date and death date. Therefore, the cohort with the highest accuracy stated above was classified as the GS cohort. The GS cohort comprised 17,735 inpatients, 497 emergency patients, 60 outpatients, and no health examination patients.

### 3.3. Comparison with the Validation Dataset

The observation period of the validation dataset was calculated as the date difference between the first hospital visit and the latest date of the visit or discharge. For the validation dataset, the mean of the observation period was 1233 days (approximately 40.5 months), and for the in-hospital deaths, the mean was 1102 days (approximately 36 months). 

The validation dataset was assessed using three classification systems: (a) Cancer stage, (b) Cancer type, and (c) Type of final visit.

#### 3.3.1. Cancer Stage

The SEER stage information was used to classify the validation dataset through the cancer stages. Each re-grouped SEER stage was plotted on a graph with a trend line for the cosine similarity values compared with the GS and mortality rate at the 2nd-level ATC class (Figure 5). The trend line had an r-squared value of 0.75. The graph and the trend line for the cosine similarity values at the 1st-level ATC class are plotted in Appendix A. 

#### 3.3.2. Cancer Type

There were 18 different types of cancer: biliary pancreatic, blood, brain tumor, breast, colorectal, esophageal, gynecological, head and neck, hepatic, lung, lymphoma, osteoblastic sarcoma, skin, stomach, thyroid, urological, and others. Graphs and trend lines for the cosine similarity values compared with the GS and mortality rates for each cancer type at the 2nd-level ATC class are plotted in Figure 6. The trend line had an r-squared value of 0.46. The graph and the trend line for the cosine similarity values at the 1st-level ATC class are plotted in Appendix A.

#### 3.3.3. Type of Final Visit

The validation dataset was categorized by final visit types. There were four types: Inpatient (I), Emergency (E), Outpatient (O), and Health Examination (G). The pharmaceutical composition names of the final administration during the four types of final visits to the GS were compared through cosine similarity. Inpatients had a cosine similarity of 0.97, with a mortality rate of 0.91. The cosine similarity of Emergency was 0.95, with a mortality rate of 0.82. Outpatient had a cosine similarity of 0.85, with a mortality rate of 0.34. Lastly, the Health Examination had a cosine similarity of 0.62, with a mortality rate of 0.01. 

## 4. Discussion

### 4.1. Main Findings

We determined that although data input errors are possible in all clinical work processes, death information is presumed to be error-prone in the manual data input process. Additionally, death information can be trusted differently based on the source, timing, and type of EHR data. Unlike measured data that can purify the error value through outlier detection, the date of death information has no criteria to confirm it even if there was a data input error. However, this study presents a measure that could determine the accuracy of the date of death information and differentiate reliability. 

### 4.2. The Negativity of Handwritten Input Compared to Automated Information Management

After the management of automated data was initiated, thus creating the EHR system, the accuracy of registered death dates improved. To accurately confirm the death date, where it could feature errors with manual data entry, specific words such as “post-mortem care” were used. This implies that the errors in death dates are not the patient’s fault but rather an administrative issue, causing a minor setback that could suggest a negative impact on data quality. Despite the recognized improvements in cost, quality performance, and job satisfaction in the change to automated data entry in hospitals [21], some areas continue to rely on manual data entry, which is error-prone. Moreover, the declaration of death is processed by a relative or co-resident of the deceased or the offices of the prefecture-owned city (Gu), center of a municipality (Eup), or village (Myeon) [22]. This is a common circumstance in which manual data entry could create errors. However, manual data entry or other errors that occur at the district offices cannot be detected. Although the death-related data of cancer patients are governed at the national level, inhomogeneous events such as pandemics could also cause a data discrepancy. In Spain, owing to the huge casualties caused by the first wave of the coronavirus pandemic (February–August 2020), the government proposed national-level objectives and homogenous criteria to maintain reliable and current information [23].

### 4.3. Comparison between the GS and Validation Dataset

Through comparison between the validation dataset and the GS, we investigated how effective the variables from the GS were as an accuracy measure of death data. First, when considering the cancer stage, Figure 5 shows that the higher the mortality rate, the higher the cosine similarity with the GS. Second, as the validation dataset was subdivided by cancer type, the higher the mortality rate, the higher the cosine similarity (Figure 6). Among the cancers, there are those with a good prognosis and those without. Regardless of the cancer type, there was a positive relationship between the mortality rate and cosine similarity. Therefore, as the GS only contains the truly dead patients, higher cosine similarity indicates a higher proportion of the truly dead patient in the dataset. As the relationship between cosine similarities for any cancer type with the mortality rate, the GS could be a successful accuracy measure of death data regardless of cancer type. Additionally, even when viewed as a type of visit, the higher the mortality rate, the higher the cosine similarity. Hence, the GS could also be used successfully as an accurate measure of any visit type in death datasets.

### 4.4. Potential of Final-Administered Medication as a Measure of Death Data

Observing the pattern of final-administered medication received by the patients who are close to death could predict death because similar conditions would require similar prescribed drugs. The most popular final prescription ATC codes used by the GS cohort were psycholeptics (N05); antihemorrhagics (B02); cough and cold preparations (R05); antibacterials for systemic use (J01); and analgesics (N02). Evidence for the popular final-administered medication ATC codes is already evident in other studies.

For comfort, psycholeptics are prescribed for patients with terminal cancer because patients experience anxiety and depression due to their stress and physical situation [24,25]. Regarding antihemorrhagics, because heart disease was the most likely non-cancer cause of death for cancer patients, either from treatment or infections stemming from the treatment process, there is a high possibility of non-cancer-related administrations [26]. This is because near-death patients with hypotension often use heart stimulants or heart-related medications among the final-administered medications, most likely because of blood pressure drops caused by multiple-organ failures, a common event that occurs in near-death patients. Further, to improve the quality of life near death for palliative cancer patients, antibiotics are used to treat infections [27], explaining the use of antibacterials and cold preparations. Moreover, regarding analgesics, during the final admission of patients with terminal cancer, opioids (fentanyl, morphine), megestrol, and metoclopramide were the most commonly used essential medications [28]. These drugs are intended to ease patients’ pain more than to treat cancer because late in the disease trajectory, there is a common trend where the goals of care transition to comfort, and referral to a hospice takes priority [29]. Through the clinical data presented, we can show that the ATC code trends of the final-administered medication from the GS are reliable due to the circumstances of near-death.

### 4.5. Strengths and Limitations

To the best of our knowledge, this is the first study to investigate the use of the final-administered medication as a measure of a cohort’s death quality using death information alone. There have been studies [3,30,31] that assess data quality through the cause of death; however, this study is the first study to assess the data quality solely on the date of death. Moreover, while there have been status reports on the quality of death information [1,2,4,5,32], this study applied the final-administered medication to further investigate the death information and its potential as a quality measure. 

As noted above, various attempts have been made to determine the quality of death data due to its implications. The methodology created in this study could assist various fields as it can verify the accuracy of death data. As the final-administered medication of GS can be an effective method to show the cancer trajectory, practitioners could comprehend patients’ cancer progress through comparisons with the final-administered medication variables of GS. Furthermore, data collectors could check dataset accuracy before proceeding with data analyses using death data. Moreover, using the final-administered medication of GS, policymakers could suggest suitable policies to assist near-death cancer patients by accessing the administered medication EHR.

This study has some limitations. Although the acquired KCCR data came from a nationwide database, the findings may not be directly applicable to other countries and settings because of the low ethnic variety in Korean ethnicity. Moreover, because the KCCR database is updated for each registered hospital every two years, some delays in death registration in recent years could exist. 

## 5. Conclusions

The difference in the accuracy of individual EHR data for death information was significant. Through numerous steps to determine the GS cohort, it was presented as patients with in-hospital deaths after 2006 with the difference between the date of death and final visit being either 0 or 1 day. Due to the introduction of automated information management, the data accuracy in in-hospital deaths since 2006 was higher compared to the deaths before 2006 and cancer-registered deaths, which contains the inaccuracy of handwritten input. Furthermore, the potential of final-administered medication as a measure of death information accuracy was found through the comparison of the cosine similarity of the GS of final-administered medication. Hence, we present the final-administered medication as a data accuracy measure of mortality data. To the best of our knowledge, this is the first study to examine the accuracy of death dates of individual EHR data and investigate the final-administered medication as a measure of a cohort’s death quality. Further improvements are possible by using other variables, such as data regarding prescriptions other than by the doctor’s order, to further determine the quality of mortality data, in addition to the final-administered medication.

## Figures and Tables

**Figure 1 cancers-15-03371-f001:**
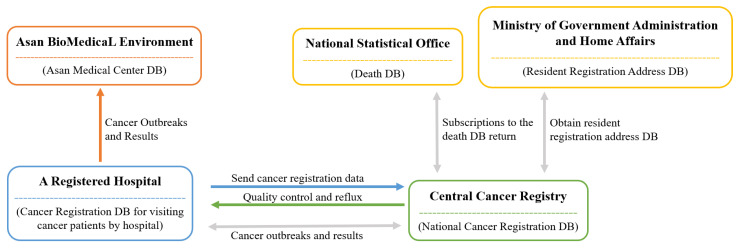
Operating systems of death databases by institution [12].

**Figure 2 cancers-15-03371-f002:**
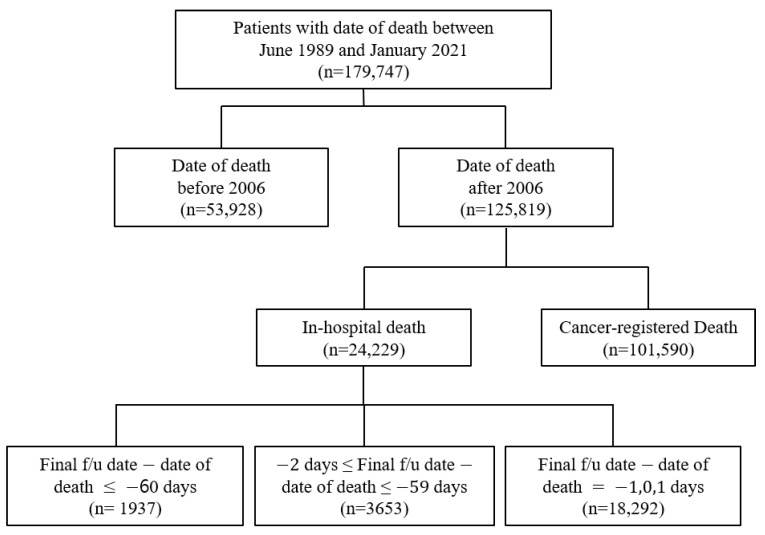
Distribution of EHR data acquired from the ABLE system.

**Figure 3 cancers-15-03371-f003:**
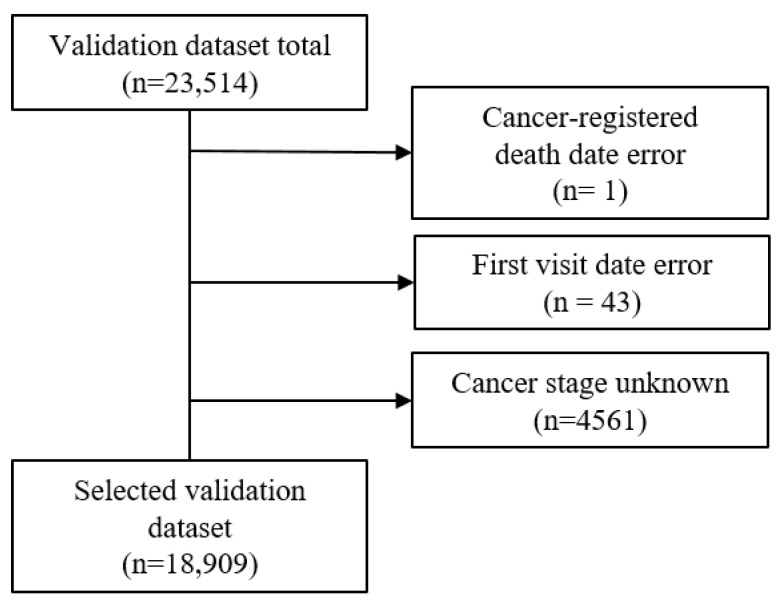
Selection of the validation dataset with the exclusion criteria.

**Figure 4 cancers-15-03371-f004:**
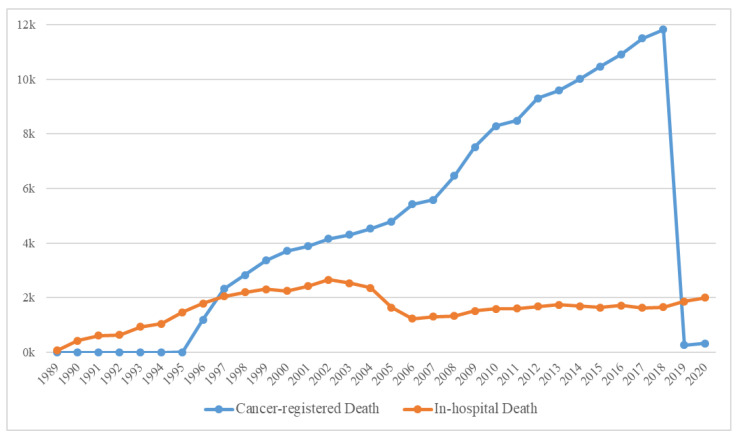
The number of in-hospital and cancer-registered deaths over time.

**Figure 5 cancers-15-03371-f005:**
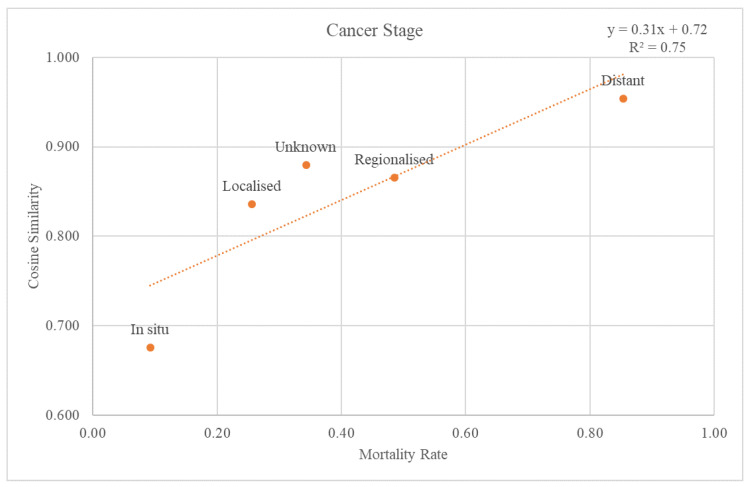
Relationship between mortality rate and regrouped cancer stages.

**Figure 6 cancers-15-03371-f006:**
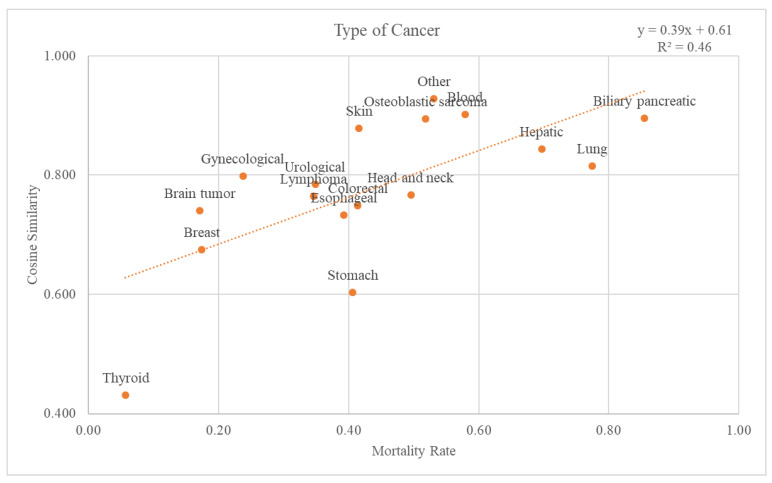
Relationship between mortality rate and cancer types.

**Table 1 cancers-15-03371-t001:** Sum of the count of each 2nd-level ATC class for group A and B.

	A01	A02	A03	A04	A05	A06	A07		V0x
Group A	$\alpha_{1}$	$\alpha_{2}$	$\alpha_{3}$	$\alpha_{4}$	$\alpha_{5}$	$\alpha_{6}$	$\alpha_{7}$	**…**	$\alpha_{n}$
Group B	$\beta_{1}$	$\beta_{2}$	$\beta_{3}$	$\beta_{4}$	$\beta_{5}$	$\beta_{6}$	$\beta_{7}$		$\beta_{n}$

## Data Availability

The datasets generated and/or analyzed during the current study are available from the institutional review board of Asan Medical Center, while restrictions apply to the availability of these data that were used under license for the current study and thus are not publicly available. The data are available from the corresponding author upon reasonable request and with the permission of the institutional review board of Asan Medical Center.

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
