# Peer review of "Examining Final-Administered Medication as a Measure of Data Quality: A Comparative Analysis of Death Data with the Central Cancer Registry in Republic of Korea"

_cancers, 2023, doi:10.3390/cancers15133371_

Round 1

Reviewer 1 Report

The issue is of interest, and the state of the art is clearly described.

The results are presented in a sufficiently clear manner, and the limitations of study are correctly mentioned. However, the methods should be clearly described, focusing on the objective of study.

In my opinion, the following points need to be better clarified:

Methods The authors did not provide information about the modality of registration of cases in Cancer Registry and the criteria of selection of cancers. Did the authors set any criteria for selection of type of cancer included in evaluation? It should more clearly described why some specific types of tumors were selected.

Moreover, were tumors staged at diagnosis? Or during the follow up? Did the authors find this information in Cancer Registry? How the data was collected?

Didn't the authors extract data on age and gender distribution?

All of these are useful pieces of information for interpreting the results and these aspects should be better specified in this section.

Results Conclusion of this study are not completely justified: I'm not completely sure that final administered medication data can be a quality indicator for death data: for example, some patients with a chronical degenerative disease received therapy for a long time. Thus, in my opinion, final administered medication data could probably be useful to increase the accuracy of death data, but not sufficiently appropriate as markers of death.

Regarding stage, did the authors analysed the correlation between stage and death according to specific tumour? The effect on death could be different according to the type of tumor, even if it was diagnosed in a metastasis stage and it would be far more interesting to add this information.

Moreover, age is an important variable and its effect on death isn’t negligible.

Author Response

Dear Reviewer,

Thank you for your thorough review of our manuscript titled Examining final administered medication as a measure of data quality: A comparative analysis of death data with the Central Cancer Registry in South Korea. We sincerely appreciate the time and effort you dedicated to providing constructive feedback. Your suggestions and comments have been instrumental in enhancing the clarity and quality of our work. We have carefully considered each of your points and would like to address them individually.

  1. Additional Explanation about KCCR:

Methods The authors did not provide information about the modality of registration of cases in Cancer Registry and the criteria of selection of cancers.

Thank you for highlighting the need for further clarification on certain aspects of our study's methodology. We have updated the manuscript as below to further clarify on the modality of registration of cases in KCCR and the criteria of selection of cancers.

Cancer cases diagnosed in hospitals nationwide during the previous year are collected in the database at the Central Cancer Registry [13]. To ensure accuracy and completeness, the KCCR collaborates with central and regional cancer registries to compile data [13]. (line 129-132)

Did the authors set any criteria for selection of type of cancer included in evaluation? It should more clearly described why some specific types of tumors were selected.

Thank you for pointing out to add criteria for selection of type of cancer included in evaluation. We have selected all the available type of cancer from the KCCR which were based on the ICD-O-3. We have updated the manuscript as below for clarity.

All cancer cases were registered based on the International Classification of Diseases for Oncology, 3rd edition (ICD-O-3) [13]. (line 135- 136)

Moreover, were tumors staged at diagnosis? Or during the follow up? Did the authors find this information in Cancer Registry? How the data was collected?

We are thankful for your suggestion for adding more information about the tumor stage specifics. We have added the following sentences below for clarification.

Moreover, the tumor stage is set within 4 months of diagnosis, as documented in the medical records, which is later captured by the KCCR [14]. These cancer-related data were obtained from the ABLE system. (line 136-139)

  1. Demographics:

Didn't the authors extract data on age and gender distribution?

We appreciate your observation that our manuscript lacked a conclusive discussion on patient demographics. We acknowledge that the primary focus of our study was the evaluation of the accuracy of medical records rather than survival analysis. However, we believed demographics table could assist readers with more understanding of the dataset, so we have included the demographics table in the supplementary table 1.Thank you for your meaningful comments

  1. Is it an indicator?

Results Conclusion of this study are not completely justified: I'm not completely sure that final administered medication data can be a quality indicator for death data: for example, some patients with a chronical degenerative disease received therapy for a long time. Thus, in my opinion, final administered medication data could probably be useful to increase the accuracy of death data, but not sufficiently appropriate as markers of death.

While we acknowledge that our current study primarily focuses on evaluating the accuracy of medical records, we genuinely believe that our methodology holds the potential to serve as an indicator in the broader context of healthcare research. The reason we used the term indicator is that as medication registered near death is readily available, therefore it will be able to evaluate the quality of the dataset efficiently. However, after consideration, we also believed the term “indicator” might be too strong, therefore has changed the word indicator to measure for clarification. We thank you for your meaningful comments.

  1. Specific Cancer Stage and Death:

Regarding stage, did the authors analysed the correlation between stage and death according to specific tumour? The effect on death could be different according to the type of tumor, even if it was diagnosed in a metastasis stage and it would be far more interesting to add this information.

We are grateful for your valuable suggestion to consider conducting a more detailed study by dividing cancer cases into specific stages and including cancer types. We recognize the importance of further research that delves into specific cancer stages, but we intended to conduct a study that could be universally applicable by including various cancer. Also, we believed the analysis was likely to be too complicated. Should we undertake future investigations, we will consider your suggestion and explore the possibility of studying specific cancer stages in more depth.

Once again, we extend our sincerest appreciation for your thoughtful review and constructive comments. We will diligently address each of your concerns in the revised version of our manuscript. Your expertise and guidance have been instrumental in improving the quality of our research.

Thank you for your continued support and valuable feedback.

Best regards,

Yae Won Tak

Reviewer 2 Report

This study assessed the quality of extracted death data in a very large dataset, collected in Korea.

I think that the topic is very original, and definitely worth investigation. One general remark: the study findings have undoubtely relevance in the Korean scenario. Do the Authors think that the study methods can be applied to different countries/areas? a more extensive comment on this will definitely increase the relevance of the paper for a general audience.

Another comment. Sorry to say, but the use of English language is quite poor, especially for the Simple Summary (which is very very very hard to understand). I appreciate that English is not Authors' first language - nor mine - but can the Authors make an effort to improve grammar and style?

See above

Author Response

Dear Reviewer

Thank you for taking the time to review my manuscript titled Examining final administered medication as a measure of data quality: A comparative analysis of death data with the Central Cancer Registry in South Korea. I appreciate your valuable feedback and comments.

Another comment. Sorry to say, but the use of English language is quite poor, especially for the Simple Summary (which is very very very hard to understand). I appreciate that English is not Authors' first language - nor mine - but can the Authors make an effort to improve grammar and style?

First and foremost, I would like to apologize for any shortcomings in the language quality of the manuscript. I understand the importance of clear and concise communication in academic writing, and I appreciate your patience and understanding in this matter. We have engaged in professional editing services to upgrade the quality of English in this manuscript. The following changes are made by the professional editing services which were double checked by the authors for their meanings.

One general remark: the study findings have undoubtely relevance in the Korean scenario. Do the Authors think that the study methods can be applied to different countries/areas? a more extensive comment on this will definitely increase the relevance of the paper for a general audience.

Also, we would like to thank you for the general remark that the study findings might be only relevant in the Korean scenario. Your suggestion to explore the expansion of our method to different geographical contexts is an attractive proposition. As we continue our research and development efforts, we will consider the feasibility and implications of extending our methods to different countries/areas. Your suggestion serves as a valuable reminder to keep an open mind and consider the broader impact of our work.

Once again, I appreciate your feedback and concerns. Your comments have helped me recognize the importance of language proficiency in academic writing, and we express our gratitude for your thoughtful suggestion and encouragement to explore our methods globally.

Thank you for your time and consideration.

Sincerely,

Yae Won Tak
